# Longitudinal Comparison of Constant Artifacts in Optical Coherence Tomography Angiography in Patients with Posterior Uveitis Compared to Healthy Subjects

**DOI:** 10.3390/jcm11185376

**Published:** 2022-09-13

**Authors:** Dominika Pohlmann, Martin Berlin, Felix Reidl, Steffen Emil Künzel, Uwe Pleyer, Antonia M. Joussen, Sibylle Winterhalter

**Affiliations:** 1Berlin Institute of Health at Charité—Universitätsmedizin Berlin, Charitéplatz 1, 10117 Berlin, Germany; 2Department of Ophthalmology, Charité—Universitätsmedizin Berlin, Corporate Member of Freie Universität Berlin, Humboldt-Universität zu Berlin, 13353 Berlin, Germany; 3Birkbeck College, University of London, Malet St., Bloomsbury, London WC1E 7HX, UK

**Keywords:** artifacts, longitudinal, optical coherence tomography angiography, uveitis posterior

## Abstract

Background: Knowledge about artifacts in optical coherence tomography angiography (OCTA) is important to avoid misinterpretations. An overview of possible artifacts in posterior uveitis provides important information for interpretations. Methods: In this monocentric prospective study, OCTA images from a total of 102 eyes of 54 patients with posterior uveitis, and an age-matched control group including 34 healthy subjects (67 eyes), were evaluated (day 0, month 3, month 6). We assigned different artifacts to distinct layers. Various types of artifacts were examined in different retinal layers. The χ2 test for the comparison between the control and uveitis group and Cochran’s Q test for the longitudinal comparison within the uveitis group were used. Results: A total of 2238 images were evaluated; 1836 from uveitis patients and 402 from healthy subjects. A total of 2193 artifacts were revealed. Projection (812 [36.3%]), segmentation (579 [25.9%]), shadowing (404 [18.1%]), and blink artifacts (297 [13.3%]) were the most common artifact types. The uveitis group displayed significantly more segmentation artifacts and projection artifacts (*p* < 0.001). No segmentation artifacts were documented in healthy subjects. The consecutive examinations within the uveitis group revealed the same artifact types without significance (*p* > 0.1). Conclusions: The uveitis patients showed more segmentation and projection artifacts than the control group. Within the uveitis group, artifacts remained longitudinally constant in terms of artifact type and pattern. The artifacts therefore appear to be reproducible on an individual level.

## 1. Introduction

Optical coherence tomography angiography (OCTA) is a new non-invasive diagnostic tool that provides detailed visualizations of different layers of the retina and choroid. The principle of OCTA is based on the different amounts of light reflected by blood cells moving through vessels. These changes in the OCT signal are detected by repeatedly capturing OCT images at each point on the retina, which yields an aggregate image that differentiates perfused vessels and static surrounding tissues. A major advancement of OCTA is that microvascular changes can be visualized with depth resolution, which makes OCTA very interesting in the field of retinology and uveitis. The use of OCTA can be advantageous in patients with ocular inflammation because it sheds light on the pathophysiology and reveals previously unseen microvascular insights. Despite the significant advantages of OCTA, different types of imaging artifacts may limit the interpretation and clinical application of this technology. Artifacts can be introduced at various stages of the imaging pipeline and can arise due to physical ocular peculiarities, movement during imaging, and errors in image processing. In general, artifacts can include structures that are not real, are missing or appear to be in the wrong place, or are represented by incorrect brightness, size, or shape [1]. Multiple studies demonstrate that the prevalence of artifacts ranges from 72 to 100% [2,3,4,5]. Therefore, awareness and recognition of artifacts—alongside knowledge of each device’s OCTA capabilities to ensure optimal use of the technology in a clinical setting—are necessary to avoid clinical misinterpretations. The correct clinical interpretation of OCTA images therefore relies on a solid understanding of retinal morphology, as well as the origin and effects of potential artifacts.

To date, only limited reports on the prevalence of OCTA artifacts among uveitis patients exist.

This study aims to determine which artifacts are prominent in posterior uveitis and whether these artifacts are reproducible.

## 2. Methods

This single-center, prospective study was performed in accordance with the Declaration of Helsinki and approved by the local ethics committee (EA4/055/16). Written informed consent was obtained from each participating patient before imaging.

In total, we examined 169 eyes of 88 patients, of which there were 102 eyes with uveitis and 67 healthy eyes, using OCTA (SPECTRALIS^®^ Heidelberg Engineering, Heidelberg, Germany) between March 2019 and September 2019. A recording of one healthy eye could not be carried out completely and was thus excluded from the evaluation. The uveitis patients were seen at three time points (day 0, month 3, and month 6) and OCTA images were taken at each visit. A 15° × 15° scan angle and high-speed mode protocol was used to acquire 261 B-scans resulting in images with an axial resolution of approximately 4 μm, B-scan resolution of approximately 11 μm, and between B-scan resolution also of approximately 11 μm. The standard OCTA viewing module (Software version 6.12.4.0) and its associated automatic segmentation of the retinal layers were applied to derive the en face slabs for each vascular plexus. 

We analyzed the following automated segmented retinal layers: the inner retinal layer that includes the superficial vascular complex (SVC) and the deep vascular complex (DVC), which is further subdivided into the superficial vascular plexus (SVP), intermediate capillary plexus (ICP), and deep capillary plexus (DCP). The outer retinal layer represents the avascular complex that normally shows no functional vessels (Figure 1).

In addition, the choriocapillaris (CC), which is located below the retinal pigment epithelium (RPE) and Bruch’s membrane (BM), of window defects was examined. 

The examination of the patients (D.P.) and the analysis of the images (M.B.) were carried out by two different persons to avoid investigator bias.

Three groups of artifacts were distinguished (Figure 2):
*(a)* *System immanent artifacts that are irrespective of the type of device:*

**projection** = vasculature structures of superficial layers are erroneously shown in deeper layers

**shadowing** (masking) = signal loss in underlying layers through density of media, such as vitreous haze

**window effect** (unmasking) = defect of the RPE (e.g., atrophy) leads to signal amplification
*(b)* *Artifacts through motions:*

**segmentation artifacts** = errors in the automatic segmentation can lead to incorrect OCTA findings

**vessel duplication** = vessels displayed twice directly next to each other
*(c)* *Artifacts through motions:*

**motion artifacts** = thin white horizontal lines resulting in an apparent disruption or displacement of the vessels

**blink artifacts** = vertical and horizontal black lines 

**banding** = adjacent horizontal stripes of different brightness

### Statistical Analysis

All data were analyzed using IBM^®^ SPSS (Armonk, NY, USA, Software version 28). The χ2 test for the comparison between the control and uveitis group and Cochran’s Q test for the longitudinal comparison within the uveitis group were used.

## 3. Results

### 3.1. Patient’s Characteristics

At our Department of Ophthalmology, 102 uveitis eyes and 67 healthy eyes of 88 patients were examined in a follow-up period of 6 months. More specifically, 64 eyes of 34 birdshot retinochoroidopathy (BSRC) patients and 38 eyes of 20 punctate inner choroidopathy (PIC) patients were enrolled. Both groups were adjusted for age and gender. The mean age was 54 (+/− 14) for the uveitis group and 53 (+/− 18) for the healthy group. The proportion of women was slightly higher in both groups (62–63%). See Table 1 for further details. 

### 3.2. Artifact Types

A total of 2238 images were evaluated: 1836 images of uveitis patients and 402 of the healthy control group. A total of 2193 (97.99%) artifacts were observed. 

The most common artifact types in all individuals were (a) projection (812 [36.3%]) and shadowing (404 [18.1%]), (b) segmentation artifacts (579 [25.9%]), and (c) blink artifacts (297 [13.3%]). 

System immanent artifacts, including projection and shadowing, were significant in both groups. When comparing the uveitis and control group, it was found that the control group had significantly more projection artifacts in the DVC and ICP (*p* < 0.001), whereas the uveitis group revealed more artifacts in DCP (*p* < 0.001). In addition, Shadowing artifacts were shown in all layers in both groups. Only a few uveitis patients showed a window effect in the CC.

The artifacts caused by data processing algorithms and image processing were particularly noticeable in all retinal layers in the uveitis group compared to the control group (*p* < 0.001). The duplication of vessels was observed in just one healthy individual. 

Motion artifacts were present in both groups, albeit in very small numbers. Blink artifacts were most prominent. (See Table 2 for further details.) 

In the follow-ups, it was found that there were no longitudinal differences over 3 or 6 months in terms of artifact type and pattern. More details are in Appendix A. 

## 4. Discussion

OCTA is an elegant method to rapidly and non-invasively visualize depth-resolved retinal and choroidal microvasculature. OCTA is growing in popularity and is an invaluable asset in the assessment of various vascular structures, not only in the diagnosis of retinal diseases, but also for uveitis. It has significant potential to diagnose and monitor distinct uveitis entities by shedding new light onto the pathophysiology of abnormal vascular changes in inflammatory conditions. As is the case for any other imaging technique, some limitations exist: image artifacts are common and can lead to misinterpretations. This highlights the necessity to identify and categorize possible artifact types and patterns, especially in certain patient groups, such as uveitis patients. Imaging artifacts have multiple causes, such as (1) system immanent artifacts, (2) artifacts from image processing, and (3) artifacts caused by movement [1]. In our study, the most common artifact types were projection artifacts (system immanent artifacts). Projection artifacts usually appear as a replication of more superficial vessels in deeper layers, occurring in the OCTA imaging process when light passes through moving blood in superficial layers before reaching and reflecting off the (deeper) target layer. The passage of light through large superficial vessels leads to OCT signal fluctuation in the deeper layers, even in the absence of erythrocyte movement. The signal fluctuates over time when the light has passed through the blood vessels, and so the reflection of this light is detected by having a decorrelation resembling blood flow. Most OCTA algorithms cannot distinguish these fluctuations from the variation of moving particles within deeper layers [6]. This results in the appearance of “false” blood flow signals in tissue regions that should be avascular, and the “false” blood vessels have the pattern of the overlying retinal blood vessels [7]. Therefore, the projection artifacts are more prominent in the deeper layers of high signal intensity. The origin of projection artifacts is concordant with our results. Moreover, projection artifacts were more present in the DVC and ICP of the healthy group. A possible explanation for the higher number of projection artifacts only in the DCP in uveitis patients could be that the light passes through the retina due to a more altered vasculature (rarefaction) until it reaches the RPE. The light on the RPE reflects towards the OCTA device. Instead of this, the vascular structure of the healthy control group is more intact in the SVP and DVP so that the reflection towards the OCTA device can develop earlier.

This limitation was known early in the inception of OCTA technology, and the development of projection artifacts removal algorithms should minimize these artifacts via post-processing. Some algorithms can mask the regions below large superficial vessels, which can present dark structures and apparently disrupt vasculature in deeper layers [8]. However, this process can lead to misinterpretations, e.g., the diagnosis of macular neovascularization (MNV), in which it is often required to examine whether new blood vessels arising from choroid are breaking through BM or the RPE to differentiate types of MNV. The outer retinal space, i.e., the avascular space between the outer nuclear layer and BM, should not contain any functional blood vessels. 

The second most common artifacts in our study were segmentation artifacts that are produced through the automatic segmentation and may lead to incorrect OCTA findings, followed by shadowing artifacts that represent signal loss in underlying layers. Compared to other studies, the prevalence of artifacts varied based on the OCTA device, the chosen settings, the type of artifacts studied, and the underlying disease [9]. Comprehensive research of the published literature, in which 59 studies were included, analyzed the prevalence of OCTA image artifacts [9]. The artifacts varied in the different studies, likely due to the different underlying diseases. In a study including patients with age-related macular degeneration (AMD), cystoid macular edema to diabetic retinopathy (DR), and retinal vein occlusion (RVO), the most common artifact was banding (89.4%), followed by segmentation (61.4%) [10]. In another study with the same underlying diseases, the most prevalent artifacts were projection artifacts (100%), segmentation error artifacts (55%), and motion artifacts (49%) [2]. Segmentation artifacts appear to be a real problem in the analysis of OCTA images. In addition, automated segmentation algorithms may cause incorrect recognition of layers [10]. In uveitis patients, the automatic segmentation is difficult and possibly incorrect due to the changed vascular structures in all retinal layers caused by retinal atrophy of the inner and outer retina or fibrotic changes. 

Manual segmentation is possible, but it is time-consuming and thus not feasible in clinical settings. Correcting segmentation is desirable, especially in longitudinal studies. In our study, the macular finding did not show any significant changes, i.e., normal macula versus macular edema. Therefore, our described artifacts were reproducible after a 3- and 6-month follow-up. 

All in all, there is a need for strategies that can further reduce artifacts. Regarding projection artifacts, projection artifact removal algorithms via post-processing exists. For example, the user of SPECTRALIS OCTA has the option to activate or deactivate the projection artifact removal to review the integrity of displayed data. These projection artifact removal algorithms minimized projection artifacts but did not remove them in our study. 

Zhang et al. proposed an algorithm to remove projection artifacts by resolving the ambiguity between in situ and projected flow signals [11]. The algorithm identifies a voxel within the in situ flow where intensity-normalized decorrelation values are higher than all shallower voxels in the same axial scan line [11]. Thus, projection artifacts suppress effectively on both en face and cross-sectional angiograms and improve the enhanced depth resolution of vascular networks [11,12]. 

The second most common artifact is the segmentation artifact that especially occurs in diseases with retinal pathologies or low-quality images [2]. SPECTRALIS OCTA provides a segmentation propagation tool to facilitate the correction of an OCTA volume. The manual correction of few scans leads to the correction of compromised slab boundaries for the entire volume [13]. Other devices´ (like the Optovue) software provides an “Edit Band/Propagation” tool which works similarly to the one offered by SPECTRALIS OCTA. 

The eradication of artifacts due to eye motion remains a technical challenge. Optovue devices utilize motion correction technology (MCT) which can improve the scan quality, but sometimes residual lines and distorted lines persist after MCT [6]. Recently, RTVue-XR systems integrated a real-time eye tracking function based on the light intensity detected by an infrared (IR) full-field fundus camera. The combination of tracking-assisted scanning with MCT registration yields a greater reduction in motion artifacts on two levels. First, eye tracking based on an IR image is performed in real-time and corrects for eye blinks, saccades, and fixation deviations. The second level is a post-processing step (MCT) that performs precise pixel level registration in three dimensions to further improve the motion correction accuracy and the resulting image quality [14]. Camino et al. demonstrated that tracking-assisted scanning, integrated with MCT, has a higher performance than tracking or MCT alone [14]. 

## 5. Conclusions

Artifacts are a common issue in OCTA imaging and can impede clinical interpretations. Projection artifacts resemble the vasculature of overlaying layers and were the most common artifact in all patient groups. Other prevalent artifacts in our study were segmentation errors, shadowing, and blink artifacts. All artifacts were reproducible over time. 

Specialists should be familiar with different types of artifacts, and we conclude that there is an emerging need to develop a grading system for OCTA imaging artifacts. 

## Figures and Tables

**Figure 1 jcm-11-05376-f001:**
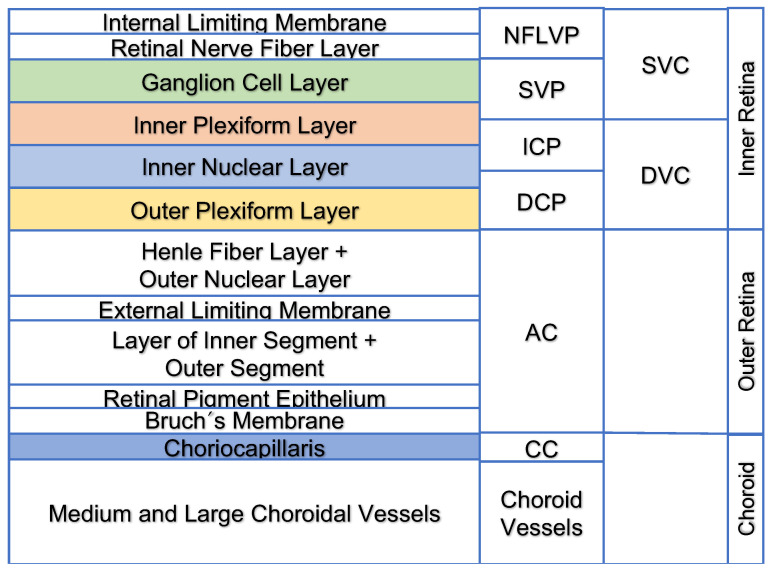
The relationship of the retina layers to the analyzed layers. (**Left**): Schematic of retinal layers from inside to outside. (**Right**): Schematic figure of the slab definitions of SPECTRALIS. Abbreviations in alphabetic order: AC: avascular complex; CC: choriocapillaris; DCP: deep capillary plexus; DVC: deep vascular complex; ICP: intermediate capillary plexus; NFLVP: nerve fiber layer vascular plexus; SVC: superficial vascular complex; SVP: superficial vascular plexus.

**Figure 2 jcm-11-05376-f002:**
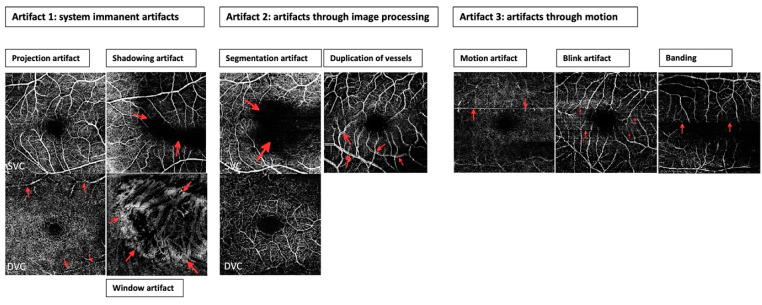
Artifact types. Three groups of artifacts can be distinguished. Artifact 1: system immanent artifacts that are irrespective of the type of device. This includes (1) projection artifacts (mostly in the deep vascular complex (DVC)): in DVC vasculature structures of superficial layers are imaged (red arrows); (2) shadowing artifacts: signal loss that image a black bar (red arrows); (3) window artifacts: visualization to choroid through retinal pigment epithelium loss (red arrows); Artifact 2: Artifacts through image processing. This includes (1) segmentation artifacts (mostly in the superficial vascular complex (SVC): errors in the automatic segmentation can conduct to unanalyzable, black areas (red arrows); (2) duplication of vessels: vessels are displayed in duplicate (red arrows). Artifacts 3: artifacts through motions. This includes (1) motion artifacts: very thin horizontal lines (red arrows); (2) blink artifacts: vertical and horizontal black lines (red arrows); (3) banding: adjacent horizontal stripes of different brightness (red arrows).

**Table 1 jcm-11-05376-t001:** Patient’s characteristics.

	Uveitis Group	Healthy Control Group
Number of patients (*n*)	54	34
Number of eyes	102	67
Age (years)		
Mean (SD), range	54 (14), 23–77	53 (18), 21–79
Sex, *n* (%)		
female	34 (63%)	21 (62%)
male	20 (37%)	12 (38%)

**Table 2 jcm-11-05376-t002:** Artifact types in uveitis and healthy group.

	Uveitis Group *n* = 102 Eyes (%)	Healthy Group *n* = 67 Eyes (%)	*p*-Value
**Artifact 1: system immanent artifacts**			
**Projection**			
SVC	0	0	1
SVP	0	0	1
DVC	84 (82.35)	67 (100)	**<0.001**
ICP	85 (83.33)	67 (100)	**<0.001**
DCP	54 (52.94)	6 (5.88)	**<0.001**
CC	0	0	1
**Shadowing**			
SVC	7 (6.86)	7 (10.45)	0.408
SVP	7 (6.86)	7 (10.45)	0.408
DVC	8 (7.84)	7 (10.45)	0.56
ICP	7 (6.86)	7 (10.45)	0.408
DCP	26 (25.49)	25 (24.51)	0.101
CC	44 (43.13)	31 (30.39)	0.689
**Window effect**			
SVC	0	0	1
SVP	0	0	1
DVC	0	0	1
ICP	0	0	1
DCP	0	0	1
CC	6 (5.88)	0	**0.043**
**Artifact 2: artifacts through image processing**			
**Segmentation**			
SVC	42 (41.18)	0	**<0.001**
SVP	42 (41.18)	0	**<0.001**
DVC	40 (39.22)	0	**<0.001**
ICP	40 (39.22)	0	**<0.001**
DCP	23 (22.55)	0	**<0.001**
CC	0	0	1
**Duplication of Vessels**			
SVC	0	1 (1.49)	0.312
SVP	0	1 (1.49)	0.312
DVC	0	0	1
ICP	0	0	1
DCP	0	0	1
CC	0	0	1
**Artifact 3: artifacts through motion**			
**Motion artifact**			
SVC	5 (4.9)	2 (2.99)	0.541
SVP	5 (4.9)	2 (2.99)	0.541
DVC	1 (0.98)	0	0.416
ICP	1 (0.98)	0	0.416
DCP	1 (0.98)	0	0.416
CC	0	0	1
**Blink artifact**			
SVC	12 (11.76)	8 (11.94)	0.972
SVP	12 (11.76)	8 (11.94)	0.972
DVC	14 (13.73)	8 (11.94)	0.736
ICP	14 (13.73)	8 (11.94)	0.736
DCP	13 (12.75)	8 (11.94)	0.877
CC	15 (14.71)	8 (11.94)	0.608
**Banding**			
SVC	2 (1.96)	0	0.249
SVP	2 (1.96)	0	0.249
DVC	2 (1.96)	0	0.249
ICP	2 (1.96)	0	0.249
DCP	2 (1.96)	0	0.249
CC	1 (0.98)	0	0.416

## Data Availability

Data available on request due to restrictions eg privacy or ethical. The data presented in this study are available on request from the corresponding author. The data are not publicly available due to privacy.

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
