# Peer review of "Longitudinal Comparison of Constant Artifacts in Optical Coherence Tomography Angiography in Patients with Posterior Uveitis Compared to Healthy Subjects"

_jcm, 2022, doi:10.3390/jcm11185376_

Round 1

Reviewer 1 Report

  • This is a prospective study evaluating differences in artifacts on OCTA imaging between uveitis patients and healthy controls. This is a useful analysis of artifacts clinicians may encounter in using OCTA, specifically in uveitis patients. OCTA is a relatively new but growing imaging study in ophthalmology. The strengths of this paper is its prospective nature and relatively large number of patients/eyes.
  • General comments:
  • Within the methods, it is not clear who is analyzing the images? Was it one person or multiple? If multiple, how were disagreements adjudicated? Additionally, were the images analyzed in a masked fashion (eg did the analyzer know if the pictures were of a uveitis patient or healthy control)?
  • Where does the term "system immanent artifacts" come from? 
  • I was not able to find a caption for Figure 1.
  •  
  • Specific comments 
  • Abstract, line 18: The sentence 'Various types were examined in different retinal layers" is unclear. What various types is this referring to?
  • Abstract, line 27: "DCP" is not defined as an acronym.
  • Methods, line 91: this sentence is not clear
  • Discussion, lines 176-183: It is not clear exactly how this post-processing is related to the identification of MNVs.
  • Discussion, lines 183-185: As this paper (Zhang et al) is discussed later in the discussion, I would consider deleting the sentence here.
  • Discussion, lines 186-187: It would be helpful to include a brief reminder of the definition of segmentation artifacts and shadowing artifacts here.
  •  

Author Response

Please look at the pfd file.

Reviewer 2 Report

The authors aims to determine which artifacts are prominent in optical coherence tomography angiography in patients with posterior uveitis compared to healthy subjects. However, there are some questions in this manuscript:

1. There are some typo and grammar errors in the manuscript. English editing is needed.

2. The authors analyzed the following automated segmented retinal layers: SVC, DVC, SVP, ICP, DCP, and CC. The authors should describe the relationship of these six layers from superficial retina to choriocapillaris in figure clearly.

3. The artifact types are very important in this manuscript. The authors should describe how to find the artifact types of OCTA in the legend of figure 1 clearly.

4. The authors should discuss the reasons why the control group had significantly more projection artifacts in DVC and ICP whereas the uveitis group revealed more projection artifacts in DCP.

5. The title "....with uveitis posterior compared...." should be "....with posterior uveitis compared.....".

6. In the conclusion of abstract, line 26: ".... segmentation, window, and projection...." should be ".... segmentation and projection....".

7. The authors should discuss the reasons why the uveitis group had significantly more segmentation artifacts in all layers except CC?

Author Response

Look at the pdf file.

Round 2

Reviewer 2 Report

The quality of this manuscript improved after revision. However, there are still some questions:

1. Abstract, Line 27: "deep capillary complex (DCP)"?

2. Legend of new Figure 1: "DCP= deep capillary complex"?

3. Figure "1". Artifact types.... should be Figure "2".

4. Legend of Figure 2:  deep vascular complex (DCP)? superficial vascular complex (SCP)? 

There are many mistakes in the segmented retinal layers as above. It makes the manuscript very confusing. Please check the segmented retinal layers and abbreviations carefully.
